# Gammaherpesvirus-infected germinal center cells express a distinct immunoglobulin repertoire

Monika A Zelazowska[1], Qiwen Dong[2,3], Joshua B Plummer[1], Yi Zhong[1], Bin Liu[1], Laurie T Krug[2], Kevin M McBride[1]

The gammaherpesviruses (γHVs), human Kaposi sarcoma-associated herpesvirus (KSHV), EBV, and murine γHV68 are prevalent infections associated with lymphocyte pathologies. After primary infection, EBV and γHV68 undergo latent expansion in germinal center (GC) B cells and persists in memory cells. The GC reaction evolves and selects antigen-specific B cells for memory development but whether γHV passively transients or manipulates this process in vivo is unknown. Using the γHV68 infection model, we analyzed the Ig repertoire of infected and uninfected GC cells from individual mice. We found that infected cells displayed the hallmarks of affinity maturation, hypermutation, and isotype switching but underwent clonal expansion. Strikingly, infected cells displayed distinct repertoire, not found in uninfected cells, with recurrent utilization of certain Ig heavy V segments including *Ighv10-1*. In a manner observed with KSHV, γHV68 infected cells also displayed lambda light chain bias. Thus, γHV68 subverts GC selection to expand in a specific B cell subset during the process that develops long-lived immunologic memory.

## Introduction

The human gammaherpesviruses (γHVs), Kaposi sarcoma-associated herpesvirus (KSHV/HHV8) and EBV (HHV4) and the murine γHV (murine γHV68/MuHV4/MHV68) are lymphotropic viruses that establish lifelong, persistent infections in hosts. They are associated with a number of neoplastic and lymphoproliferative diseases (Cesarman, 2014), particularly upon immunosuppression (Damania & Munz, 2019). In addition, there is an association with certain autoimmune diseases, including multiple sclerosis (Almohmeed et al, 2013; Ascherio & Munger, 2015). MHV68 serves as a mouse model by virtue of its structural and genetic homology to human γHVs and pathology resembling that of primary EBV infection in humans (Speck & Virgin, 1999; Tarakanova et al, 2005; Barton et al, 2011; Cieniewicz et al, 2016; Dong et al, 2017).

In mice, MHV68 intranasal (IN) infection results in acute viral replication and amplification in mucosal tissue, migration to secondary lymphoid organs, expansion in germinal center (GC) B cells, and ultimately long-term latency in memory B cells. Within the GCs, MHV68 is thought to be predominantly latent, with low viral replication and gene expression (Barton et al, 2011). Although a number of lymphoid and CD11c[+] myeloid populations are associated with the spread of the virus to secondary lymphoid organs (Gillet et al, 2015), how γHV gains access to GC cells in vivo is not known. Because γHV can infect various B cell types (Barton et al, 2011; Collins & Speck, 2012; Dong et al, 2017), direct infection of GCs is a potential route. Conversely, γHV may infect non-GC cells and reprogram them into a GC-like cell. Because B cells express unique immunoglobulin receptors, the antigen specificity and origin of the infected GCs could be consequential to both the host and virus.

Activated B cells enter the GC to undergo affinity maturation, the process by which B cells evolve improved antibody affinity. The process is driven by somatic hypermutation (SHM) and follicular helper (Tfh) T-cell selection (Victora & Nussenzweig, 2012; De Silva & Klein, 2015). During SHM, random mutations are introduced into variable regions (Di Noia & Neuberger, 2007). Mutations that increase B-cell receptor (BCR) affinity provide a selective advantage for antigen interaction and uptake. This, in turn, facilitates Tfh interaction, continued survival, and clonal expansion. Cells that successfully undergo affinity maturation differentiate into long-lived antibody-secreting plasma cells and memory B cells (Shlomchik & Weisel, 2012). Thus, accessing GCs would give MHV68 a pathway to memory and plasma cell compartments (Flano et al, 2002). BCR crosslinking can reactivate the γHV lytic programs (Moser et al, 2005; Kati et al, 2013), thus inhabiting antigen-sensitive cells could impact latency establishment and reactivation from memory. Viral reprogramming of irrelevant cells into GC cells would abrogate antigen influence but leave the quandary of how selection and survival of these cells would occur in the GC.

γHVs encode numerous genes that can manipulate cell signaling, survival, and proliferation (Coleman et al, 2014; Price & Luftig, 2014; Sin et al, 2015; Williams et al, 2015; Romero-Masters et al, 2018; Bravo

[1]Department of Epigenetics and Molecular Carcinogenesis, Science Park, The University of Texas MD Anderson Cancer Center, Smithville, TX, USA  [2]Department of Molecular Genetics and Microbiology, Stony Brook University, Stony Brook, NY, USA  [3]Graduate Program of Molecular and Cellular Biology, Stony Brook University, Stony Brook, NY, USA

Correspondence: kmcbride@mdanderson.org
Qiwen Dong's present address is Department of Medicine-Infectious Disease, University of Chicago, Chicago, IL, USA
Laurie T Krug's present address is Center for Cancer Research, National Cancer Institute, Bethesda, MD, USA

Cruz & Damania, 2019). Ex vivo infection models have shown that EBV and KSHV can induce activation-induced cytidine deaminase expression, affect hypermutation, and alter class-switch recombination (CSR) (Bekerman et al, 2013; Kalchschmidt et al, 2016; Rosario et al, 2018). Furthermore, KSHV infection of cultured lymphocytes induces RAG1/2 protein expression, receptor editing, and a shift from *IGK* to *IGL* surface expression (Totonchy et al, 2018). While aberrant V(D)J recombination, CSR, and SHM promote lymphomagenesis, altered selection can hinder antibody response and induce autoimmunity (Alt et al, 2013; Nemazee, 2017; Kuraoka et al, 2018), and the mechanistic details of how γHVs impact antibody diversification and repertoire selection during latent GC expansion in vivo remain poorly defined.

To investigate the dynamic between the virus and host GC cells, we analyzed the GC repertoire from MHV68 infected mice. We used the transgenic virus, MHV68-H2BYFP, which expresses histone H2B fused to EYFP fluorescent protein to identify infected GC B cells in vivo (Collins & Speck, 2012). Mouse studies demonstrate that with both IN and intraperitoneal (IP) inoculation, acute viral replication is cleared and the peak latency occurs 14–18 days postinfection (dpi). At this point, most MHV68+ cells are latent GCs cells (Collins & Speck, 2012). We find that these MHV68+ GCs express a distinct Ig repertoire, not found in the uninfected GC pool of cells, and provide the first in vivo evidence that the virus actively subverts the GC selection process.

# Results

### Tracking MHV68 in the GC

To understand how GC repertoire is affected by a γHV in the context of the initial colonization of the lymphoid tissue (or during the establishment of latency), we established a protocol to analyze individual MHV68+ cells from the GC population of infected mice. To determine the dynamics of GC and MHV68+ cell expansion during infection, we infected mice with 1,000 PFUs of MHV68-H2BYFP via either IN or IP inoculation. At 14, 16, and 18 dpi, splenocytes were evaluated by flow cytometry (Fig S1), and the relative percentage of GC (CD19[+], GL7[+], and CD95[+]) (Fig 1A) or YFP[+] of total B cell (CD19[+], CD4[−], and CD8[−]) populations was determined (Fig 1B). The GC compartment was found to be significantly expanded 14–16 dpi with the kinetics of IN inoculated mice slightly delayed compared with IP-inoculated mice. YFP[+] cells were detected at day 14 with peak expansion observed between 16 and 18 dpi (Fig 1B). More than 60% of YFP[+] were GC with 2–10% of total GCs being YFP[+] (Fig S1). Similar to previously reported GC dynamics during MHV68 infection (Collins & Speck, 2012), we found significant GC expansion and YFP presence. Thus, we demonstrated the ability to identify in vivo, MHV68-infected GCs cells via their associated YFP[+] signal in vivo.

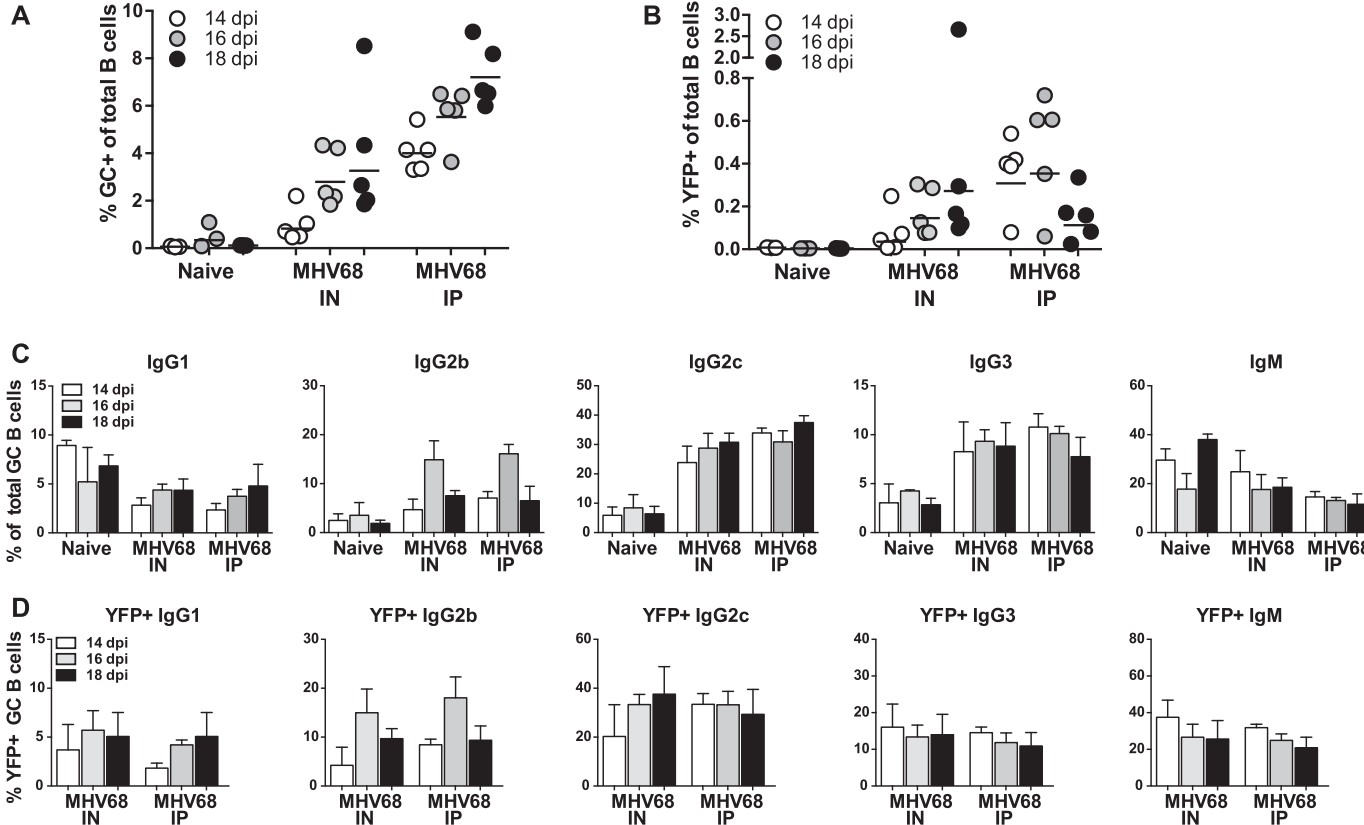

**Figure 1. Dynamics of B cells in MHV68-H2BYFP–infected mice.**
**(A)** Flow cytometry analysis of germinal center (GC) cells (CD19[+], GL7[+], and CD95[+]) as a percentage of total spleen B cells. Each circle is the analysis of an individual mouse 14, 16, or 18 days postinfection (dpi) via intranasal (IN) or intraperitoneal (IP) MHV68-H2BYFP inoculation. Naïve, uninfected mice were used as control. **(B)** Summary of YFP[+] (MHV68-YFP[+]) cells as a percentage of total splenic B cells. **(C, D)** Isotype expression profile of total GC B cells or (D) YFP[+] GC B cells from the spleen of control naïve, IN, or IP inoculated mice at the indicated dpi.

We investigated how MHV68 infection affects isotype switching by measuring the isotype expressed by GC cells from the spleens of naïve and infected mice. GC cells from infected mice displayed a shift towards IgG2b, IgG2c, and IgG3 isotypes with a drop in IgG1 and IgM (Figs 1C and S1). This shift was noted with both inoculation routes and was evident 14–18 dpi. Analysis of the MHV68+ population demonstrated a similar distribution of isotype expression (Fig 1D), suggesting that isotype expression in the MHV68+ population is driven by the overall GC response and host response to infection.

### MHV68+ GC cells express lambda light chain

The GC reaction drives selection of B cells expressing higher affinity BCRs through the affinity maturation process. To determine if infected GC cells experience the same selective process, we compared the repertoire of MHV68+ (YFP+) and MHV68− (YFP−) from the same mice. GC cells were isolated by single-cell FACS from the spleens of IN inoculated mice 17 dpi. At this time point, the MHV68+ population in the spleen has undergone significant expansion (Fig 1). A total of five samples were processed for analysis: Sample 1 was from two pooled animals, whereas samples 2 through 5 were from individual mice. RT-PCR of the expressed variable region Ig heavy ($V_H$), kappa ($V_K$), and lambda ($V_L$) variable regions were performed on individual cells from each sample. Amplification efficiency was similar between populations with a total of 440 $V_H$ and 525 $V_{KorL}$ sequences obtained from MHV68+ cells, with 443 $V_H$ and 490 $V_{KorL}$ sequences from MHV68− cells (Table S1).

Analysis of light chain expression revealed that infected cells displayed a significant shift in the *Igl* usage ($P < 0.001$, two tailed *t* test) (Fig 2). Whereas most MHV68− GC cells express *Igk*, MHV68+ cells expressed *Igk* or *Igl* light chains at nearly equal frequency. This trend was significant when considering the entire population of light chains (Fig S2) or only cells where matched heavy and light chains were recovered (Table S1 and Fig 2). Dual $\kappa/\lambda$ light chain expression, a result of the receptor editing process, does occur normally in a low percentage of mature B cells (Rezanka et al, 2005; Luning Prak et al, 2011). We found cells that expressed both $\kappa/\lambda$ light chain in a small percentage and not significantly different between the populations.

### Ighv and Ighj usage in MHV68+ B cells

We next determined if the divergence observed in light chain use was also reflected in the heavy chain repertoire. In mature B cells, the expressed *Igh* variable domain is a recombined exon consisting of single variable (V), diversity (D), and joining (J) gene segments, which were assembled during B cell development in the bone marrow (Jung et al, 2006; Schatz & Swanson, 2011). Within the variable domains, three complementarity-determining region (CDR) loops constitute the antigen-binding paratopes (Sela-Culang et al, 2013), with CDR1 and CDR2 encoded entirely within V segments and CDR3 coded by the V(D)J junction. Therefore, which *Ighv* gene encodes the V segment can be a major determinant of antibody–antigen binding properties. We analyzed productive *Igh* sequences from 329 MHV68+ and 360 MHV68− GC cells (Table S1) for *Ighv* usage. We identified a total of 64 *Ighv* gene segments expressed in all populations. 48 *Ighv* exons

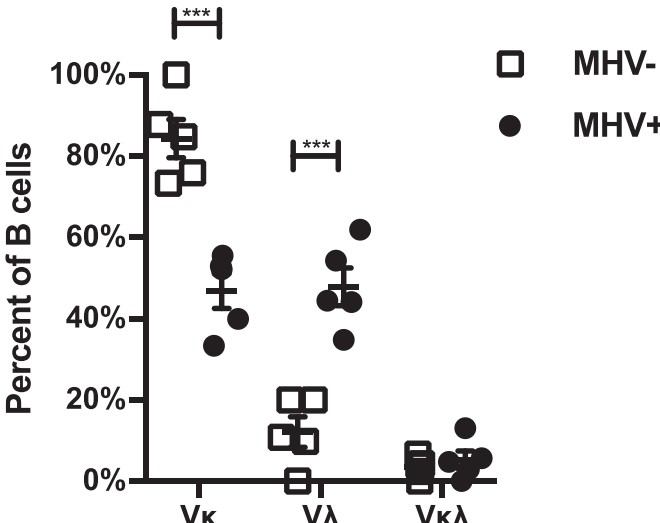

**Figure 2. MHV68-infected germinal center cells express lambda light chain more frequently.**
Graph shows the percentage of cells expressing Kappa (Vκ), Lambda (Vλ), or both light chains from cells where the matching heavy chain was also identified. Single MHV+ and MHV− germinal center cells were isolated from the spleens of intranasal MHV68-inoculated mice 17 days postinfection on the basis of YFP+ or YFP− expression. Analysis of expression was by RT-PCR and sequencing of individual cells. Five independent biological replicates were analyzed with replicate sample 1 consisting of two pooled mice and samples 2 through 5 from individual mice. The mean and SEM are displayed. *t* test, ***$P < 0.001$.

were found in the MHV68+ population, whereas 56 were found in MHV68−. 8 *Ighv* exons were exclusively present in MHV68+, whereas 16 (25%) were unique to MHV68− cells. This finding suggests the MHV68+ population has a more confined *Ighv* repertoire.

*Ighv* genes are grouped into families based on shared evolutionary and sequence identity. We find a significant difference in the frequency that *Ighv1* and *Ighv10* family genes were used. Usage of *Ighv1*, the largest family representing ~60% of all *Ighv* genes, was significantly lower in the MHV68+ population (Fig 3A). In contrast, the *Ighv10* family was significantly higher in the MHV68+ population. Comparing utilization of individual *Ighv* genes, we repeatedly observed a distinct bias towards specific *Ighv* genes in each population. Most dramatic was *Ighv10-1* in the MHV68+ population, which was highly used in all samples (~20%) and the top *Ighv* in three of five samples (Fig 3B). In contrast, *Ighv1-82* was the most frequently expressed *Ighv* exon in 4 of 5 MHV68− samples but was rare or absent in the MHV68+ populations. In addition, we observed sporadic high abundance of a particular *Ighv* (i.e., *Ighv7-3*) in samples of the MHV68+ population. D gene segments are very short, often altered by the recombination and hypermutation, and difficult to identify with certainty. Therefore, we next focused on J segments. Of the four *Ighj* gene segments expressed in the mouse, there was a significant bias towards *Ighj4* in the MHV68+ population (Fig 3C). In summary, we observed a significant divergence of *Ighv* usage between MHV68+ and MHV68− populations with *Ighv10-1* overrepresented in MHV68+ cells. The fact that samples 1–3 were independently obtained from mice that we housed and processed at a different facility (see the Materials and Methods section) than samples 4 and 5 further supports that this bias is driven by the virus.

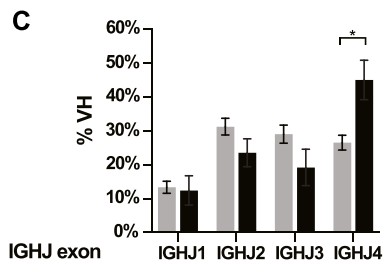

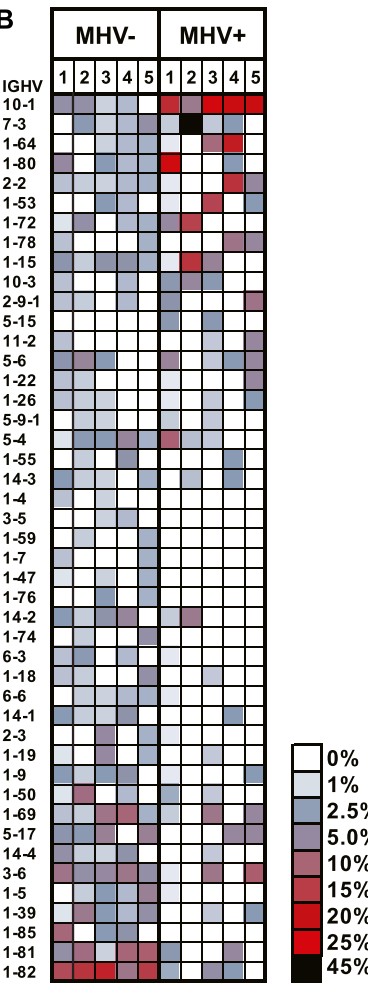

**Figure 3. Differential usage of Ighv and Ighj exons in MHV68+ and MHV68− germinal center cells.**
**(A)** Percentage of MHV+ and MHV− germinal center B cells that express an *Ighv* gene belonging to the indicated family. The mean of five independent biological sample is displayed. **(B)** Heat map of relative *Ighv* gene usage of MHV+ and MHV− populations from each of biological samples. Only *Ighv* segments present in more than one sample are displayed. Rows are ordered from the greatest positive (top) to negative (bottom) mean differential between the MHV+ and MHV− populations. **(C)** Summary of relative *Ighj* exon usage. Graphs show mean values from the five biological replicate samples with SEM. *t* test, *P ≤ 0.05, **P ≤ 0.01, ***P ≤ 0.001.

## Clonal expansion in MHV68+ B cells

γHV are known to cause proliferation and expansion of infected B cells. To determine how MHV68 affects clonal selection and evolution in the GC, we compared clonal presence and expansion of the GC MHV68+ and MHV68− populations. Within each sample, we grouped sequences into clonal clusters based on their *Ighv*, *Ighj*, and CDR3 length and the presence of N-1 nucleotides (V, J, [AA]). In total, 161 clonal groups were found in MHV68+ (individual samples: 66, 10, 35, 23, and 27) and 331 in MHV68− GC cells (individual samples: 105, 62, 69, 51, and 44).

In the MHV68+ population, there was expansion of clonal subsets with a few specific clones constituting a significant percentage of each sample (Fig 4A). In contrast, the MHV68− population was highly diverse with only 6.9% of clonal groups containing more than one sequence. In three of the five samples, the most abundant MHV68+ clone expressed *Ighv10-1*. Furthermore, *Ighv10-1* is used by multiple clones in each group. In total, there are 26 clonal groups with *Ighv10-1* in MHV68+, and 12 of them consist of more than one sequence, whereas all 10 clones with *Ighv10-1* in MHV68− contain only one sequence each. Comparison of clonal groups between MHV68+ and MHV68− population revealed little (0–5%) overlap (Fig 4B). In all five

samples, only 14 clonal groups were present in both populations and these were of variable size (Fig 4C). Taken together, these results indicate that the repertoire of the MHV68+ GC cells is largely distinct from the rest of the GC repertoire.

## Hypermutation characteristics

To determine if MHV68 impacts SHM in cells, we investigated the frequency and location of mutations in GC cells. A region of the expressed *Ighv* gene downstream of our amplification primer and upstream of the CDR3 region encompassing CDR1 through framework 3 (FR3) regions was compared with the germline sequence. We included only sequences with sufficient quality to definitively call mutations. Both populations had a similar distribution of mutated clones, with most sequences displaying between 0 and 2 mutations in the analyzed *Ighv* region (Fig 5A). Overall, the MHV68+ population displayed a small (20%), but statistically significant (unpaired *t* test, *P* = 0.01) increase in mutations (Fig 5A). However, the degree of difference was variable from sample to sample (Fig S3). The location of mutations in both populations were biased to the CDR regions, a phenomenon known to be driven by clonal selection (Fig 5B). Since the detected mutations could have existed before MHV68 infection,

**A**

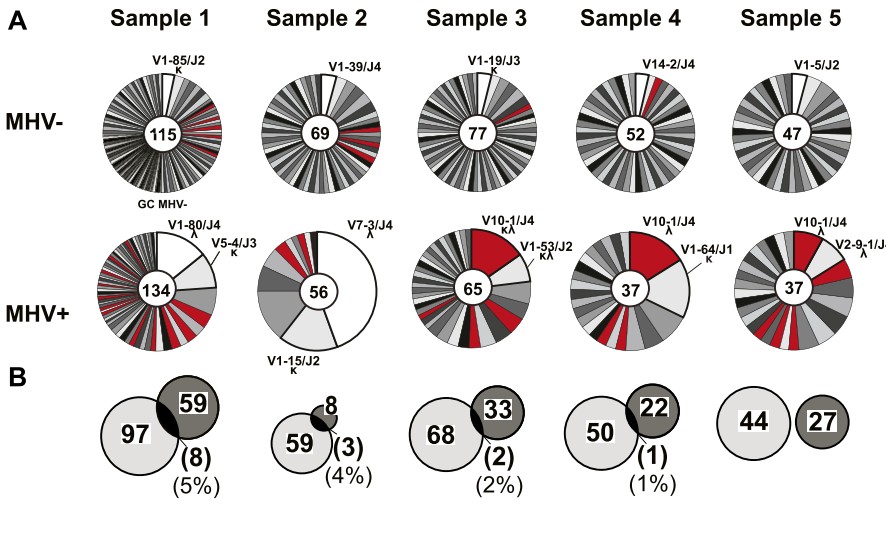

**B**

**C**

Figure 4. Clonal expansion of distinct repertoire in MHV68+ germinal center cells.
**(A)** Pie charts show proportion of individual cells belonging to distinct clonal groups within each population from five samples. Size of slice is proportional to number of sequences in each clonal group defined as sharing common *Ighv*, *Ighj*, and V(D)J junction. Number of total cells included in the pie chart is indicated in the center of each chart and the top clonal group(s) is(are) listed by *Ighv*, *Ighj*, and light chain type (κ or λ). Clonal groups that use *Ighv10-1* are colored (Red). **(B)** Clonal groups with cells in both populations. Venn diagram circles are proportional to the number of clonal groups (listed inside circle) within MHV+ and MHV− populations of each sample. Brackets indicate the number of and relative percentage of clonal groups that were found in both populations. **(C)** Graph shows the number of cells that belong to the 14 clonal groups that were present in both the MHV+ and MHV− germinal center populations.

we analyzed the phylogenic mutation profile in several expanded clonal groups (Fig 5C). This analysis revealed that ongoing somatic mutation and isotype switching occurred in MHV68+ expanded clonal groups. Although CDR3 length distribution was more heterogeneous in the MHV68+ population due to expansion of individual clones, the difference in average length (11.6 versus 11.5) was not significantly different (*P* = 0.68, unpaired *t* test). Overall, there was a small but statistically significant difference in CDR3 average charge (MHV68+, −0.12 versus MHV68−, −0.33, unpaired *t* test *P* = 0.02) (Fig 5D and E).

### High-throughput sequencing data

To determine if our single cell analysis was indicative of overall repertoire bias in the whole GC compartment, we performed high-throughput sequencing of the GC *Igh* repertoire. Three C57Bl/6 mice were infected with MHV68-H2BYFP via IP inoculation and 20,000 MHV68+ and MHV68− GC cells were isolated 16 dpi from each spleen. The *Igh* variable region was amplified by RT-PCR and sequenced by MiSeq paired-end 300-bp sequencing. We identified 90, 82, and 82 expressed *Ighv* exons in the three samples of MHV68− cells and 78, 66, and 59, respectively, in MHV68+ cells. This reflected the trend of the more confined MHV68+ *Ighv* usage observed in the single cell analysis. Sequences were classified into clonal groups, based on *Ighv*, *Ighj*, CDR3 length, and a CDR3 identity of a least 85% (Hershberg & Luning Prak, 2015; Wu et al, 2016). For the three samples, we identified 7,623, 6,449, and 6,872 clonal groups in MHV68− populations and 4,429, 2,128, and 2,358, respectively, for the MHV68+ populations (Table S2). Our amplification method did not allow for quantitative determination

of the size of each clonal group; therefore, we assessed the relative overall percentage of clonal groups that expressed a particular *Ighv* gene (Fig 6A). Just as in our single cell analysis, there was a divergence in the *Ighv* usage between the populations. *Ighv10-1* was the first or second most frequently used *Ighv* in clonal groups from the MHV68+ population and *Ighv1-82* was the most frequent *Ighv* in the MHV68− population. There was also a higher frequency of clonal groups with *Ighj4* in the MHV68+ population (Fig 6B). CDR3 length was similar between populations with average amino acid length for each sample (MHV68+, 11.8, 11.1, and 11.1 AA; MHV68−, 11.1, 10.9, and 11.4 AA), although MHV68+ length distribution was the irregular reflecting expansion of specific clones (Fig 6C). Thus, these high-throughput results of unchanged CDR3 length but highly skewed V gene usage of MHV68+ strongly corroborate the single cell analysis (Figs 3C and 5D).

### Clonal overlap between MHV68-infected and noninfected GC cells

Our single cell repertoire analysis (Fig 4B) demonstrated little clonal overlap between MHV68− and MHV68+ GC populations. We, therefore, investigated if this was also the case for our high-throughput analysis of the GC compartment. We first evaluated whether our approach was suitable to detect clonal overlap in samples. As a control, two replicate samples of 20,000 GC cells were sorted from the same control mouse spleen. The *Igh* repertoire was RT-PCR–amplified and sequenced by high-throughput sequencing. In the two control populations, we identified 14,166 and 12,106 clonal groups, of which 8,865 were present in both. Thus, our method detected a high proportion of clonal overlap in two populations

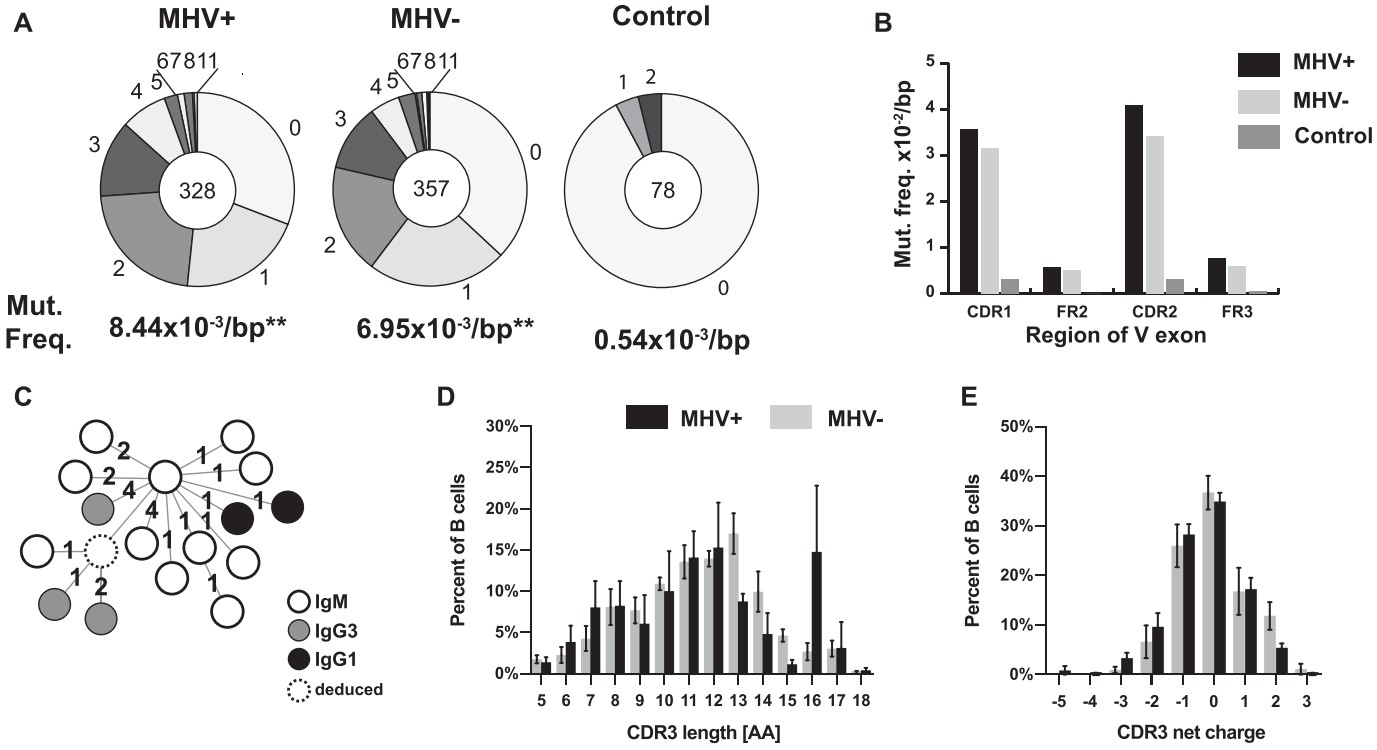

**Figure 5. Comparison of hypermutation and CDR3 characteristics between MHV68+ and MHV68− germinal center (GC) cells.**
**(A)** Pie charts show number of mutations in the expressed *Ighv* exon (from CDR1 to FR3) from MHV+ and MHV− GC cells as well as non-GC IgM⁺ control group from all samples. Pie chart wedges are proportional to the number of sequences with the indicated number of mutations. Total number of sequences analyzed from five samples is displayed in the center and average mutation frequency is indicated as mutations per base pair (bp) **P ≤ 0.01. **(B)** Mutation frequencies per bp calculated separately for indicated CDR and framework (FR) regions of the *Ighv* exon. **(C)** Representative MHV+ clone Ig phylogenic tree. Circle indicate a node (clonally related sequence) with isotype shown by the color. Dashed circle represents hypothetical split node intermediate. Each number denotes the number of mutations between nodes. **(D, E)** Analysis of relative length and (E) net charge distribution of the CDR3 region from the MHV+ and MHV− populations. The CDR3 characteristics from individual cells of all five samples is displayed as mean and SEM.

when it was expected (Fig 6D). Comparison of clonal groups in the MHV68− and MHV68+ populations found that only 7.2–11.1% were present in both populations (Fig 6D). Of the clonally related sequences present in both populations, we sought to determine whether the relative expansion of the MHV68+ population was related to the size of the group in the MHV68− population. To do this, we compared the relative number of reads as a proxy for the size of each group. The control experiment demonstrated that read numbers generally correlated between the replicant samples, especially in groups with higher read numbers (Fig 6E). However, the number of reads from the MHV68+ did not correlate with the those from the MHV68− population. Thus, this high-throughput analysis, together with single cell data, reveals a striking lack of clonal overlap between MHV68+ and MHV68−. Considering that MHV68+ cells displayed highly reproducible bias for *Ighv10-1*, hypermutation, and evidence of antigen driven clonal evolution, we conclude that MHV68 subverted infected cells to undergo an abnormal GC selection process.

### Antibody reactivity

To determine if MHV68+ and MHV68− GC B cell express immunoglobulins reactive against self- and/or viral antigens, we cloned and recombinantly expressed antibodies. A representative set of

matching heavy and light chains were expressed recombinantly in HEK293 cells and purified from media. To determine specificity against nuclear antigens, an anti-nuclear antibody (ANA) ELISA was performed (Fig 7A). None of the antibodies from either the MHV+ or MHV− displayed anti-ANA reactivity. Viral reactivity was assessed by staining MHV68-infected NIH3T12 cells and analyzed by flow cytometry (Figs 7B and S4). Staining with anti-myc–negative control antibody was used to set background. A positive control antibody against ORF46 (MHV UNG) demonstrated that viral antigens could be detected in a proportion of YFP⁺ infected cells. There was a significant difference in antiviral reactivity between the populations. Whereas only one antibody (6%) from the MHV+ population was reactive to MHV68-infected NIH3T12 cells, a significantly higher percentage (6 of 17, 35%) of antibodies from the MHV− population did (Figs 7B and S4). None of the antibodies stained noninfected NIH3T12 cells above background levels (data not shown).

## Discussion

During primary infection, gammaherpesvirus-infected cells appear in the GC, a step thought to facilitate latent presence in the memory B cell compartment. Our analysis of MHV68+ GC cells show they

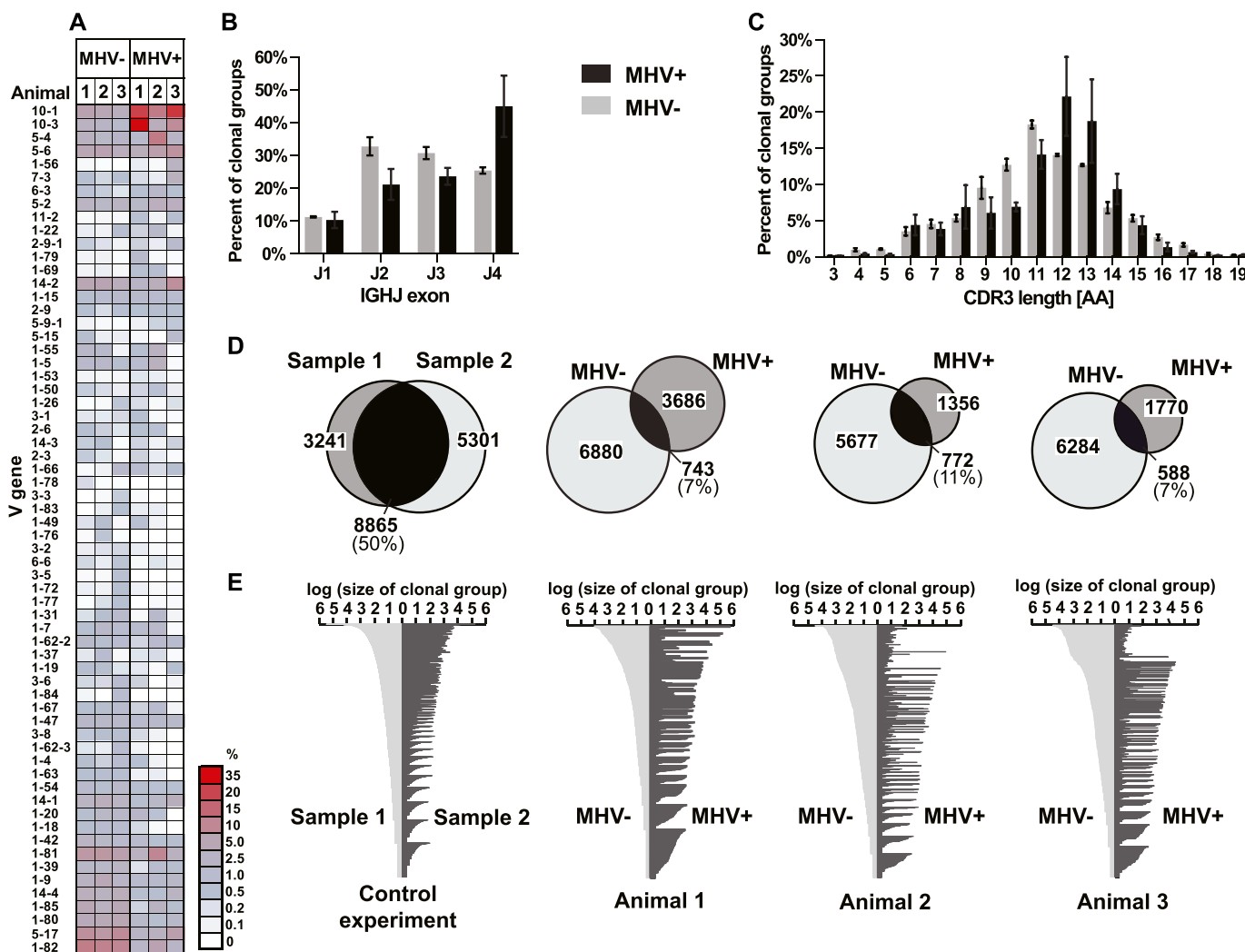

**Figure 6. High-throughput sequencing repertoire and clonal analysis.**
**(A)** Heat map showing the percentage of clonal groups using specific *Ighv* segments. Each sample consists of 20,000 MHV+ or MHV− germinal cells from individual mouse spleens 16 d after IP MHV68 inoculation. Rows are arranged from the greatest positive to negative mean differential between MHV+ and MHV− populations. *Ighv* segments are displayed that were present in more than one sample with a frequency above 0.1%. **(B)** Percentage of clonal groups using specific *Ighj* segments. **(C)** Percentage of clonal groups with designated *Igh* CDR3 length. Graphs show mean and SEM from three samples. **(D)** Clonal groups with the presence in both populations. Venn diagram circles are proportional to the number of clonal groups (listed inside circle) within MHV+ and MHV− populations of each sample. Control samples are replicate germinal center samples from a control mouse spleen. **(E)** Graphs display clonal groups with sequences MHV+ and MHV− populations. Number of reads from the MHV+ and MHV− populations are displayed for each clonal group.

carry distinct Ig repertoire with very little overlap in MHV68− GC cells. This was apparent in both the Ig heavy and Ig light chain sequences, demonstrating a novel means by which gamma-herpesvirus subverts normal B cell selection. For the light chain repertoire, MHV68-infected cells displayed a remarkable bias towards lambda usage. Because the ability of MHV68 to infect B cells independent of BCR specificity has been well established (Barton et al, 2011; Decalf et al, 2014; Frederico et al, 2014; Rekow et al, 2016), receptor editing is likely driving expression of lambda. During B cell development, light chain rearrangement is sequential, with *Igl* used only if *Igk* rearrangements fail to produce a functional, non-autoreactive BCR (Brauninger et al, 2001). In wild-type mice under steady-state conditions, the vast majority of B cells are kappa+ (Haughton et al, 1978). During the process of tolerance, B cells with

self-reactivity can induce additional light chain rearrangement, including the *Igl* locus. Consistent with this, we found a clonal *Igh* population with individual cells differentially expressing *Igk* or *Igl*. In humans, a bias towards lambda in KSHV-infected cells has been a long-known phenomenon (Du et al, 2001; Hassman et al, 2011), and cell culture studies suggest that KSHV does so by directly inducing RAG proteins and receptor editing (Totonchy et al, 2018). This suggests that like KSHV, receptor editing is shaping the light chain repertoire of MHV68-infected cells in vivo. Induction of receptor editing and lambda expression would potentially alter Ab specificity. Our analysis of recombinant antibodies from the MHV+ population revealed no increase in ANA reactivity. Although the MHV− population expressed many BCRs with clear anti-viral reactivity (35%), there was a significant drop in the percentage of the

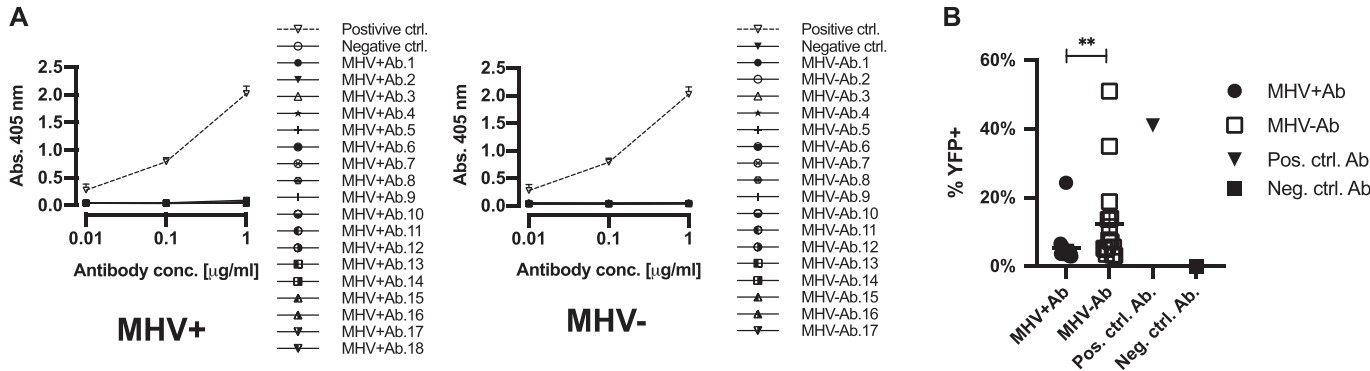

**Figure 7. Reactivity of immunoglobulins to self and viral antigens.**
**(A)** ELISA test for reactivity against nuclear antigens. Graphs show $OD_{405}$ values at 0.01, 0.1, and 0.01 µg/ml antibody concentrations of recombinant antibodies from MHV+ and MHV− populations. Anti-DNA antibody-positive control BV17-45 (dashed line) and negative control anti-myc antibody are displayed. The mean and SEM are displayed for each antibody tested in duplicates. **(B)** Antiviral specificity of antibodies. Flow cytometry analysis of cells infected with MHV68 and stained with recombinant antibodies. Plot shows percent of MHV infected (YFP[+]) cells reactive with individual recombinant antibodies from indicated populations. Positive control antibody that recognizes ORF46 (MHV68UNG) and negative control anti-myc are indicated. Mann–Whitney test, **$P < 0.01$.

MHV− population (6%). This could be due to viral selection of a nonreactive population. Alternatively, viral induction of receptor editing may have changed the original BCR's specificity.

Secondary rearrangements on light chain could indicate that similar processes were occurring on heavy chain. Secondary V(D)J combination that replaces the V segment could modify antibody specificity (Zhang et al, 2004; Sun et al, 2015). Although thought to be uncommon in mature B cells, this mechanism can be used to rescue unproductive/out-of-frame V(D)J rearrangements or to eliminate autoreactive repertoire during early B development (Chen et al, 1997; Lutz et al, 2006). The presence of $V_H$ replacement product is detected in the $V_H$–$D_H$ junction (P and N nucleotides), where a short stretch of original *Ighv* gene is left as a footprint. Sequences of these footprints indicate if $V_H$ replacement took place and which *Ighv* gene was exchanged during secondary rearrangement. Analysis of the $V_H$–$D_H$ junction region of immunoglobulins from MHV68+ cells did not show any difference in the length of N and P sequences between infected and noninfected cells. Moreover, the sequences in V–D junctions did not display the short stretch of nucleotides that are a replacement footprint. Thus, there was no clear evidence that MHV68-infected B cells use $V_H$ replacement to diversity their repertoire.

A surprising finding of this study is the confined use of *Ighv* segments and recurrent selection of *Ighv10-1* by infected MHV68+ GCs. The $Igh^b$ haplotype of C57Bl/6 has ~100 functional *Ighv* genes (Johnston et al, 2006). The *Ighv10* family is one of the smallest in $Igh^b$, consisting of two *Ighv* germline genes, 10-1 and 10-3 (Whitcomb et al, 1999). Single cell analysis demonstrated a significant increase of 10-1 in both the percentage of clonal groups expressed and expansion of groups with *Ighv10-1* (Figs 3B and 4A). Although our high-throughput sequencing analysis could not quantitatively assess clonal expansion, there was also significant preference of clonal groups that use 10-1 (Fig 6A). Recent literature analyzing *Ighv* usage does not describe unexpected abundance of 10-1 in naïve and other B cell compartments (Collins et al, 2015; Prohaska et al, 2018), although one study reported slightly higher utilization in splenic B-1a (Yang et al, 2015). Selection of 10-1 in MHV68+ cells was observed in mice purchased from different

vendors and housed at different universities and thus is a quality intrinsic to the host–pathogen interaction.

Interestingly, the *Ighv10* family has been suggested to have intrinsic DNA-binding properties and may be overrepresented in autoreactive antibodies (Chao & Voss, 1992; Jang & Stollar, 2003; Maranhao et al, 2013). Structural analysis revealed that residues coded by the *Ighv10* gene segment rather than CDR3 structure or accompanied light chain can interact with DNA. In particular, a cationic binding pocket formed by CDR2 has a role (Maranhao et al, 2013). Several autoantibodies where 10-1 participates in binding have also been described (Brigido & Stollar, 1991; Brigido et al, 1993; Li et al, 2001). In humans, EBV has been found in self-reactive cells (Tracy et al, 2012). For example, selective use of *IGHV4-34*, an *IGHV* with self-reactive properties, has been found in certain cases of infectious mononucleosis (Bhat et al, 2004; Mockridge et al, 2004) and EBV-linked chronic lymphocytic leukemia (Kostareli et al, 2009) suggesting a connection between *IGHV* usage and gammaherpesvirus pathogenesis in humans. In mice, early establishment of MHV68 infection has been noted to coincide with transient induction of sera self-reactivity (Darrah et al, 2019).

The distinct *Igh-V* repertoire in YFP-positive cells indicates that MHV68 is not infecting existing GC cells. Previous analysis with transgenic mice expressing hen egg lysozyme (HEL[+])–specific B cells also supports this concept (Decalf et al, 2014). Although MHV68 could infect HEL[+] B cells in vitro, in mice MHV68 was excluded from the HEL[+] fraction of GC cells even with HEL antigen stimulation. We posit that MHV68 induces irrelevant cells to join the GC in a noncanonical fashion. The GC is an open structure that allows circulating B cells from the follicles to move through them. Immunologically, this arrangement allows rapid scanning for antigen-specific cells and inclusion into the existing GC (Victora & Nussenzweig, 2012). Therefore, infected marginal zone (MZ), B1, and naïve B cells that circulate through the follicle could all potentially join GCs. MZ B cells are infected by MHV68 (Marques et al, 2003) and have been implicated in trafficking MHV68 to GCs as mice without MZ cells are deficient in MHV68 GC colonization (Frederico et al, 2014). MZ is a heterogeneous population with subsets that develop both within and outside the GC (Song & Cerny, 2003; Cerutti

et al, 2013) and so would be a candidate pool. How infected cells would proliferate in the GC is not clear because BCR antigen affinity and signals from T cells are required. One possibility is that the virus will induce the GC program in cells independent of antigen affinity. Alternatively, MHV68 may induce survival in cells normally eliminated, such as self- or irrelevant-antigens.

The viral infection caused a notable increase in IgG2b-, IgG2c-, and IgG3-expressing GC cells overall. However, within the infected mice, there was little difference in isotype distribution between the MHV68+ and MHV68– populations. The MHV68+ cells were very similar to MHV68– in both the mutation frequency and bias of location towards CDR loops. Thus, although the GC processes of SHM and CSR appear grossly normal in MHV68+ cells, it was clear from expanded MHV68+ clonal groups that mutations were accumulating post infection. Altogether, the mutation frequency and CDR preference, confined *Ighv* use, and bias for *Ighj4* suggest that antigen–BCR interaction is influencing MHV68+ cells. EBV-infected peripheral blood from human patients also displays hypermutation with hallmarks of antigen selection (Tracy et al, 2012). The subversion of the B-cell antigen selection process by the virus highlights a novel means to imprint potentially pathologic repertoire into the immune memory. Ex vivo analysis demonstrated BCR engagement can trigger reactivation of latent cells (Moser et al, 2005). How in vivo antigen selection and BCR engagement impacts maintenance of chronic infection and viral pathogenic outcomes, such as lymphomas, remain to be understood.

# Materials and Methods

### Mice

Wild-type C57BL/6 mice were purchased from Harlan/Envigo RMS and bred at the Stony Brook University Division of Laboratory Animal Research facility or purchased from Jackson Laboratories and bred at The University of Texas MD Anderson Cancer Center, Science Park. All animal protocols were approved by the Institutional Animal Care and Use Committee of Stony Brook University and The University of Texas MD Anderson Cancer Center.

### Infection and flow cytometry

The recombinant MHV68-H2BYFP bacterial artificial chromosome was kindly provided by Samuel Speck (Collins & Speck, 2012). 8–16-wk-old mice were inoculated with 1,000 PFUs of MHV68-H2BYFP via IN route in 20 $\mu$l of media (DMEM supplemented with 10% FBS, pen/strep and L-glutamine) or by IP in 0.5 ml media (Dong et al, 2018). The inoculum dose was validated by plaque assay titration. For FACS, spleens were harvested 16 dpi (IP) or 17 dpi (IN). Single cell suspensions of splenocytes were resuspended in PBS supplemented with 2% FBS. Fc receptors were blocked (TruStain fcX clone 93; BioLegend) before incubation with antibodies. Infected GC B cells (YFP$^+$ GC$^+$) were gated on YFP$^+$ B cells (CD19$^+$ CD4$^-$ CD8$^-$ YFP$^+$) and then GC cells (CD95$^+$ GL7$^+$). Uninfected GC B cells (YFP$^-$GC$^+$) were gated on YFP$^-$ B cells (CD19$^+$ CD4$^-$ CD8$^-$ YFP$^-$) and then GC cells (CD95$^+$ GL7$^+$) and uninfected non-GC B cells (YFP$^-$ GC$^-$) were gated on

YFP$^-$ B cells (CD19$^+$ CD4$^-$ CD8$^-$ YFP$^-$) and then non-GC cells (CD95$^-$GL7$^-$). For high-throughput analysis via next generation sequencing, 20,000 cells were collected in DMEM supplemented with 10% FBS, centrifuged at 600$g$, and washed with PBS before RNA preparation. Post-sort purity was 94–99%. For single cell sorting, the cells were deposited directly into empty wells of 96-well plates and stored at –80°C until processing. The samples were sorted using a BD FACSARIA Fusion (BD Biosciences). For isotype expression analysis, spleens were harvested at 14, 16, and 18 dpi and single cell suspensions of splenocytes were generated. Fc receptors were blocked and the cells were stained with markers for T cells (CD4, CD8), B cells (CD19), and isotype classes (IgD, IgG1, IgG2b, IgG2c, IgG3, and IgM). Data were acquired on an LSR Fortessa (BD Biosciences) and analyzed using FlowJo 10.4 (Treestar). Statistical analysis was performed using GraphPad Prism (v.6). For the control mouse immunization, mice were immunized with 50 $\mu$g of an irrelevant (GFP) recombinant protein in alum via IP administration 10 d before harvest.

### Antibodies

CD4 clone GK1.5; CD8 clone 53-6.7; CD19 clone 6D5; CD95 clone Jo2; GL7 clone GL7; IgD clone 11-26c.2a; IgG1 clone RMG1-1; IgG2b clone R12-3; IgG2c clone RMG2a-62; IgG3 clone R40-82; and IgM clone 11/41. All antibodies were purchased from BioLegend except antibodies against CD95, IgG2b, IgG3, IgM (BD Pharmingen), and GL7 (eBiosciences).

### Immunoglobulin amplification and sequencing

To amplify expressed Ig genes from single cells, cDNA was synthesized as described previously with the addition of immunoglobulin gene-specific primers (Table S3) (Tiller et al, 2009). Amplification was of the variable regions of heavy (*Igh*) and light (*Igk* and *Igl*) by nested PCR (Table S3). The amplification reaction was set up on 384-well plates and cycled as follows: *Igh*: 94°C for 1 min, 25 cycles of 94°C for 15 s, 56°C for 15 s, and 68°C for 15 s; *Igk*: 94°C for 1 min, 40 cycles of 94°C for 15 s, 61°C–50°C (1°C/s), and 68°C for 15 s; *Igl*: 94°C for 1 min, 35 cycles of 94°C for 15 s, 58°C for 15 s, and 68°C for 15 s. Second round PCR: *Igh*: 94°C for 1 min, 50 cycles of 94°C for 15 s, 60°C for 15 s, and 68°C for 15 s; *Igk*: 94°C for 1 min, 30 cycles of 94°C for 15 s, 61°C–50°C (1°C/s), and 68°C for 15 s; *Igl*: 94°C for 1 min, 35 cycles of 94°C for 15 s, 58°C for 15 s, and 68°C for 15 s. Individual amplicons were Sanger sequenced. For high-throughput sequencing of B cells sorted in bulk (20,000 cells/pellet), *Igh* amplifications were performed as above with the following alterations. PCR round 1 was 19 cycles and round 2 was 38 cycles. Each sample was amplified in 14 separate reactions and the final PCR products were pooled, purified by gel-extraction, and sequenced at The University of Texas MD Anderson Cancer Center and Science Park Next Generation Sequencing Core on the Illumina MiSeq using paired-end read sequencing (2 × 300 bp).

### Ig single cell sequence and antibody analysis

Ig-heavy and Ig-light chain sequences from single cells were analyzed manually in 4Peaks (Nucleobytes) and low-quality (average PHRED quality <30) sequences were disregarded. High-quality

sequences were converted into the FASTA format and analyzed with ImMunoGeneTics (IMGT)/V-Quest (Giudicelli & Lefranc, 2011) to identify the germline V, D, and J segments. CDR3 length was determined by counting residues from conserved cysteine position 104 according to IMGT numbering to the conserved tryptophan–glycine motif in *Ighj* segments or conserved phenylalanine–glycine in *Igkj* and *Iglj segments*. Only sequences that coded for a productive Ig were considered. The Vh segment from CDR1 to FR3 was analyzed for mutations, and the frequency was calculated as the number of nucleotide mutations over the total length of analyzed region. *Igh-V* sequences were assigned into clonal groups manually on the basis of identical *Ighv* and *Ighj*, CDR3 length, and sequence similarity of N and P nucleotides. To produce recombinant antibodies, amplified Ig variable regions were cloned into mammalian expression vectors carrying mouse Igγ1, Igκ, Igλ1, or Igλ2 constant regions and expressed in HEK293 cells (Tiller et al, 2009). Antibodies were purified, concentration determined by 280 nm absorbance, and confirmed by SDS–PAGE and Coomassie staining. To determine anti-nuclear binding properties of the recombinant antibodies, we used the ANA Screen 8 pooled antigen ELISA kit (IBL/Tecan). Antibody stocks were diluted to 0.01, 0.1, and 1 µg/ml. ELISA plates were read with an Omega FLUOstar Omega plate reader (BMG Labtech) at 405 nm. Recombinant BV17-45 antibody was included as an anti-DNA positive control and recombinant anti-myc (9e10) as a negative control (Smith & Voss, 1990). To analyze reactivity against viral antigens, NIH3T12 cells were infected with MHV68-H2BYFP (4,000 PFUs/ml media). 48 h after infection, the cells were trypsinized and washed with PBS with 1% FBS. The cells were fixed with 4% paraformaldehyde in PBS and permeabilized with 0.1% Triton X-100. Fixed cells were incubated with recombinant Abs (1 µg/ml in 1% BSA in TBS-T) for 30 min. A recombinant antibody against MHV68 ORF46 was used as a positive control. Anti-mouse IgG1 Alexa 647 was used for antibody detection. Data was acquired on LSR Fortessa (BD Biosciences) and analyzed using FlowJo 10.4 (Treestar).

### High-throughput sequence analysis

Vh paired-end reads from high-throughput sequencing were merged using PEAR (*P*-value < 0.0001) (Zhang et al, 2014). Sequences were then de-multiplexed and collapsed with cutadapt and FASTX-Toolkit. Reads were submitted to IMGT/HighV-Quest (Li et al, 2013) for annotation of V, D, and J segments. Only productive reads with no stop codons and in-frame junction (as defined by IMGT) were used for further analysis. CDR3 was defined as for single cell analysis (from codon $C_{104}$ to $W_{118}$ according to IMGT numbering). Clonal clustering was performed as in Galson et al (2015). In brief, all sequences that share the same V and J gene and the CDR3 length were grouped into bins. D gene was omitted because of the inherent inaccuracy of annotation. Next, CDR3s within each bin were compared with each other and clustered into clonal groups on the basis of amino acid sequence similarity. Each clonal cluster was characterized by "cluster center" (Galson et al, 2015), i.e., CDR3 AA sequence that is ≥85% identical with all other CDR3 in that group (Greiff et al, 2015; Hershberg & Luning Prak, 2015; Wu et al, 2016; Meng et al, 2017). Clonal groups that contained only one sequence were omitted. Clonal overlap was defined as the number of clonal groups that contained two or more sequences from both the MHV68+ and MHV68– populations.

# Supplementary Information

# Acknowledgements

We acknowledge Laurie Levine in Stony Brook University (SBU) Division of Laboratory Animal Resources for assistance with husbandry, and the SBU flow cytometry facility. We thank Samuel Speck for providing the MHV68-H2BYFP bacterial artificial chromosome. This work was supported by National Institutes of Health (NIH) AI125397 and AI111129 and the Three Strohm Sisters Family Foundation. We acknowledge the University of Texas MD Anderson Cancer Center (UTMDACC) Research Animal Support Facility (P30 NIH CA16672) and Center for Cancer Epigenetics. Cancer Prevention and Research Institute of Texas (CPRIT) Core Facility Support Grants to UTMDACC flow cytometry and cell imaging core (CPRIT RP170628), UTMDACC Science Park Next Generation Sequencing Core (CPRIT RP120348 and CPRIT RP170002) and UTMDACC Recombinant antibody production core (CPRIT RP190507).

## Author Contributions

MA Zelazowska: conceptualization, formal analysis, investigation, and methodology.
Q Dong: formal analysis, investigation, and methodology.
JB Plummer: formal analysis, investigation, and methodology.
Y Zhong: resources, software, formal analysis, and methodology.
B Liu: resources, software, formal analysis, and methodology.
LT Krug: conceptualization, resources, formal analysis, investigation, and methodology.
KM McBride: conceptualization, formal analysis, investigation, and methodology.

### Conflict of Interest Statement

The authors declare that they have no conflict of interest.

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
