## [Reviewer comments · Life Science Alliance]

Gammaherpesvirus Infected Germinal Center Cells Express A Distinct Immunoglobulin Repertoire

Monika Zelazowska, Qiwen Dong, Joshua Plummer, Yi Zhong, Bin Liu,
Laurie Krug, and Kevin McBride
DOI: 10.26508/lsa.201900526

*Corresponding author(s): Kevin McBride, The University of Texas MD
Anderson Cancer Center*

Review timeline:

Submission Date:	2019-08-19
Editorial Decision:	2019-09-13
Revision Received:	2020-01-03
Editorial Decision:	2020-01-27
Revision Received:	2020-01-28
Accepted:	2020-01-29

Scientific Editor: Andrea Leibfried

Transaction Report:

September 13, 2019

Re: Life Science Alliance manuscript #LSA-2019-00526

Prof. Kevin M McBride
The University of Texas MD Anderson Cancer Center
Epigenetics and Molecular Carcinogenesis
1808 Park Road 1 C
Smithville, TX 78957

Dear Dr. McBride,

Thank you for submitting your manuscript entitled "Gammaherpesvirus Infected Germinal Center Cells Express A Distinct Immunoglobulin Repertoire" to Life Science Alliance. The manuscript was assessed by expert reviewers, whose comments are appended to this letter.

As you will see, the reviewers are supportive of publication of your manuscript here, and provide constructive comments that will allow you to further strengthen your work. We would thus like to invite you to submit a revised version of your manuscript, addressing the comments made by rev#2 and #3.

Thank you for this interesting contribution to Life Science Alliance. We are looking forward to receiving your revised manuscript.

Sincerely,

Andrea Leibfried, PhD
Executive Editor
Life Science Alliance
Meyerhofstr. 1
69117 Heidelberg, Germany

t +49 6221 8891 502
e a.leibfried@life-science-alliance.org
www.life-science-alliance.org

B. MANUSCRIPT ORGANIZATION AND FORMATTING:

Reviewer #1 (Comments to the Authors (Required)):

Zelazowska et al. assess the impact of gHV infection on GC B cell repertoires using an elegant MHV68 infection model, which allows tracking of infected cells by viral EYFP expressing. By flow cytometry the authors assess the GC response at cellular level over the infection and compare the Ig repertoire of infected and non-infected GC B cells at single-cell level in individual mice. Their comprehensive analysis of a large number of several hundred paired Ig gene sequences provides clear evidence that MHV-infected and non-infected GC B cells in the same animals express different Ig genes. Light chain isotype and VH usage, SHM load, and clone analyses provide independent evidence. Lambda light chains, Ighv 10-1 and Ighj 4 genes are overrepresented in infected cells, which also frequently belong to expanded and diversified cell clusters with high numbers of SHM compared to their non-infected counterparts. Confirmation for the strong clonal expansion and lack of clonal overlap between infected and non-infected cells comes from bulk analyses. It seems that the

differences between the infected and non-infected cells may be even larger if the authors had performed clonal suppression analyses and normalizations to count for the differences in sampling depth. Entropy analyses might be helpful, however, none of these analyses are required given the strong differences that the authors observe. As is, the data is highly valuable and of great relevance for our understanding of the role of HV infections in biasing B cell repertoires.

The data is of high quality, the experimental approaches are well described, and the manuscript is well written. The authors present a careful and well-justified interpretation of their findings in the context of the available literature. Therefore, the manuscript merits publication without any changes.

Reviewer #2 (Comments to the Authors (Required)):

Manuscript Nr: LSA-2019-00526

Zelazowska et al., "Gammaherpesvirus Infected Germinal Center Cells Express A Distinct Immunoglobulin Repertoire"

The authors demonstrate that murine gamma herpesvirus 68 (MHV68) infection in mice skews their B cell repertoire. As previously reported for the human gamma herpesvirus Kaposi sarcoma associated herpesvirus (KSHV) MHV68 infected cells are enriched for lambda light chain usage. Furthermore, also their variable region usage in the BCR heavy chain is skewed, while isotype and somatic hypermutation characteristics do not differ from uninfected germinal center B cells in the same animals. From these data the authors speculate that MHV68 could shape the specificity of their infected B cells, possibly eliciting autoreactivities that could support the survival of their host cells.

These are interesting findings, but no information is provided on the infected B cell specificities. Some information on the possibly autoimmune or virus specific nature of the infected B cell specificities would significantly strengthen the novelty of the submitted study.

Major comments:

1. The authors report BCR modulation in MHV68 infected B cells. It would be interesting if MHV68 could directly induce the respective machinery, as has been previously reported for the Epstein Barr virus (EBV). Along these lines the authors should investigate if AID and RAG expression is induced upon in vitro mouse B cell infection by MHV68?
2. From the reported data it remains unclear what the function of BCR heavy chain variable region and lambda light chain enrichment in MHV68 infected B cells could be. Would sorted infected B cells depending on their BCR specificity survive better or worse upon adoptive transfer into non-infected mice?
3. Are any of the enriched BCRs of MHV68 infected B cells virus specific or autoreactive? The authors speculate that MHV68 aims for an alteration in the BCR specificity of its infected B cells. The novelty of the submitted manuscript could be significantly increased if some of the BCR specificities by recombinantly expressing them as antibodies could be checked for autoreactivity and virus specificity.

Minor comments:

1. Since the authors found a lambda lc shift with MHV68 in this study that is reminiscent of KSHV, I would suggest that they rather emphasize the similarities with KSHV in the abstract. This would also be more consistent with the subfamily of gamma-herpesviruses to which both MHV68 and KSHV belong.

In summary, the authors describe interesting alterations in the BCR repertoire of

MHV68 infected B cells. In order to interpret these findings, however, some more information on the function of the resulting immunoglobulins should be provided.

Reviewer #3 (Comments to the Authors (Required)):

Human gamma-herpesviruses like EBV and KSHV establish latent infection in memory B cells. It is well established that they reach this compartment by driving infected cells through GC reactions. Less is known how the virus exploits GC reactions. This study addresses this issue, in particular, an outstanding question, that is, the BCR specificity of the infected cells. The study is very well performed and data of high quality with overlapping methodologies that produce same observations and conclusions. Specifically, it is shown that: 1) there is a similar distribution of isotype expression in MHV68 infected CG B cells indicating GC driven response; 2) infected GC B cells display a shift towards IgI usage with Igk paired similar frequency; 3) significantly small clonal overlap between infected versus non-infected GC B cells; 4) somatic hypermutation in infected GC B cells. Together, these data show that MHV68 infected GC B cells are a specific B subset and is coherent with previous studies demonstrating that infection of GC B cells is not a stochastic event and not driven by ongoing humoral responses to viral antigens. The authors raise the hypothesis of MZ B cells being a possible seeding candidate. I think that is the most likely explanation. In this context the first study showing MZ B cells infection should be cited (J Virol. 2003 Jul;77(13):7308-18). It would have been significant if this study had addressed the antigen specificity of infected B cells, namely, the possibility discussed of auto-antigen reactivity. Hence, BCR specificity of infected B cells remains elusive. Interesting to note that given that EBV infects virtually all human population, evolution pressure to infect a restricted BCR repertoire that do not induce pathology is very likely.

We thank the editor and reviewers for their effort and detailed critique on our manuscript. In response to reviewer #2 & #3 comments we include new data (figure 7 and S4). Changes in the updated manuscript are highlighted. Reviewer comments and our detailed response (**in Bold**) are included below.

All the reviewers made supportive comments.

Reviewer #1: "The authors present a careful and well-justified interpretation of their findings in the context of the available literature. Therefore, the manuscript merits publication without any changes."

Reviewer #2 was also supportive stating "These are interesting findings" with the suggestion, "Some information on the possibly autoimmune or virus specific nature of the infected B cell specificities would significantly strengthen the novelty of the submitted study."

Reviewer #3: "This study addresses this issue, in particular, an outstanding question, that is, the BCR specificity of the infected cells. The study is very well performed and data of high quality with overlapping methodologies that produce same observations and conclusions." Included was the comment that "antigen specificity of infected B cells, namely, the possibility discussed of auto-antigen reactivity" would increase the significance.

As detailed below we address a question of antigen specificity of the BCR from infected cells. This is included as new data in additional Figure 7 and S4. We have now recombinantly produced antibodies and analyzed reactivity to self-nuclear and viral antigens from both MHV+ and MHV- populations. While find no increase in ANA reactivity, we find a significant drop in anti-viral reactivity of antibodies from the MHV+ population. We feel this additional data addresses the key comments by reviewers and increases the significance of this study.

Reviewer #2 (Comments to the Authors (Required)):

Manuscript Nr: LSA-2019-00526

Zelazowska et al., "Gammaherpesvirus Infected Germinal Center Cells Express A Distinct Immunoglobulin Repertoire"

The authors demonstrate that murine gamma herpesvirus 68 (MHV68) infection in mice skews their B cell repertoire. As previously reported for the human gamma herpesvirus Kaposi sarcoma associated herpesvirus (KSHV) MHV68 infected cells are enriched for lambda light chain usage. Furthermore, also their variable region usage in the BCR heavy chain is skewed, while isotype and somatic hypermutation characteristics do not differ from uninfected germinal center B cells in the same animals. From these data the authors speculate that MHV68 could shape the specificity of their infected B cells, possibly eliciting autoreactivities that could support the survival of their host cells.

These are interesting findings, but no information is provided on the infected B cell specificities. Some information on the possibly autoimmune or virus specific nature of the infected B cell specificities would significantly strengthen the novelty of the submitted study.

We address this point in major comment #3 below and include new data (New Figures 7 and S4).

Major comments:

1. The authors report BCR modulation in MHV68 infected B cells. It would be interesting if MHV68 could directly induce the respective machinery, as has been previously reported for the Epstein Barr virus (EBV). Along these lines the authors should investigate if AID and RAG expression is induced upon in vitro mouse B cell infection by MHV68?

We agree that this is an interesting point. We attempted to address this issue, as suggested by the reviewer, by infecting primary B cells in vitro. Although the primary cells infected with a good efficiency, infected cells rapidly began dying. We posit that the MHV68 infection of B cells in vitro induced an almost complete lytic program. We feel that measuring AID and RAG1/2 expression in such dying cells would not be physiologically relevant. We are also analyzing gene expression from infected germinal center cells in vivo. Preliminary results do not indicate an upregulation in AID and RAG1/2 however additional analysis is needed that will take significant time and effort.

2. From the reported data it remains unclear what the function of BCR heavy chain variable region and lambda light chain enrichment in MHV68 infected B cells could be. Would sorted infected B cells depending on their BCR specificity survive better or worse upon adoptive transfer into non-infected mice?

We agree this is a very interesting point. However this study did not conduct an analysis of survival and we feel this is beyond the scope of this current study.

3. Are any of the enriched BCRs of MHV68 infected B cells virus specific or autoreactive? The authors speculate that MHV68 aims for an alteration in the BCR specificity of its infected B cells. The novelty of the submitted manuscript could be significantly increased if some of the BCR specificities by recombinantly expressing them as antibodies could be checked for autoreactivity and virus specificity.

Thank you for commenting, this was an excellent suggestion that reviewer 3 also mentioned. We address this point with new data included in New Figure 7 and S4. To determine if BCRs from the analyzed cells were reactive to self or viral antigens we cloned and produced a representative number of antibodies from the infected and non-infected populations. Using an ANA ELISA assay, we found no significant gain of self-reactivity. We assessed anti-viral reactivity by staining 3T12 cells that were either mock treated or infected and analyzing by flow cytometry. We found that the percentage of antibodies that were

reactive against virus-infected cells was significantly lower in the MHV+ population than the MHV-.

Minor comments:

1. Since the authors found a lambda lc shift with MHV68 in this study that is reminiscent of KSHV, I would suggest that they rather emphasize the similarities with KSHV in the abstract. This would also be more consistent with the subfamily of gamma-herpesviruses to which both MHV68 and KSHV belong.

Thank you for pointing out the Rhadinovirus parallels and overlap of Lambda LC shift with KSHV. Since we focused on germinal center cells we drew parallels to EBV. We now also mention KSHV in the abstract.

In summary, the authors describe interesting alterations in the BCR repertoire of MHV68 infected B cells. In order to interpret these findings, however, some more information on the function of the resulting immunoglobulins should be provided.

As mentioned above, we address this point with new data included in New Figure 7 and S4.

Reviewer #3 (Comments to the Authors (Required)):

Human gamma-herpesviruses like EBV and KSHV establish latent infection in memory B cells. It is well established that they reach this compartment by driving infected cells through GC reactions. Less is known how the virus exploits GC reactions. This study addresses this issue, in particular, an outstanding question, that is, the BCR specificity of the infected cells. The study is very well performed and data of high quality with overlapping methodologies that produce same observations and conclusions. Specifically, it is shown that: 1) there is a similar distribution of isotype expression in MHV68 infected CG B cells indicating GC driven response; 2) infected GC B cells display a shift towards Igl usage with Igl paired similar frequency; 3) significantly small clonal overlap between infected versus non-infected GC B cells; 4) somatic hypermutation in infected GC B cells. Together, these data show that MHV68 infected GC B cells are a specific B subset and is coherent with previous studies demonstrating that infection of GC B cells is not a stochastic event and not driven by ongoing humoral responses to viral antigens. The authors raise the hypothesis of MZ B cells being a possible seeding candidate. I think that is the most likely explanation. In this context the first study showing MZ B cells infection should be cited (J Virol. 2003 Jul;77(13):7308-18).

Thank you for pointing out the reference omission. We now include this reference

It would have been significant if this study had addressed the antigen specificity of infected B cells, namely, the possibility discussed of auto-antigen reactivity. Hence, BCR specificity of infected B cells remains elusive. Interesting to note that given that EBV infects virtually all human population, evolution pressure to infect a restricted BCR repertoire that do not induce pathology is very likely.

As also mentioned above in response to reviewer two. We address this point with new data included in New Figure 7 and S4. To determine if BCRs from the analyzed cells were reactive to self or viral antigens we cloned and produced a representative number of antibodies from the infected and non-infected populations. Using an ANA ELISA assay, we found no significant gain of self-reactivity. We assessed anti-viral reactivity by staining 3T12 cells that were either mock treated or infected and analyzing by flow cytometry. We found that the percentage of antibodies that were reactive against virus-infected cells was actually significantly lower in the MHV+ population than the MHV-.

2nd Editorial Decision

27 January 2020

January 27, 2020

RE: Life Science Alliance Manuscript #LSA-2019-00526R

Prof. Kevin M McBride
The University of Texas MD Anderson Cancer Center
Epigenetics and Molecular Carcinogenesis
1808 Park Road 1 C
Smithville, TX 78957

Dear Dr. McBride,

Thank you for submitting your revised manuscript entitled "Gammaherpesvirus Infected Germinal Center Cells Express A Distinct Immunoglobulin Repertoire". Two of the original reviewers re-assessed your work, and we would be happy to publish your paper in Life Science Alliance pending final minor revisions:

- Please address reviewer #3's remaining concerns via text changes
- Please upload the supplementary tables and include a short legend for them in the main manuscript file

A. FINAL FILES:

B. MANUSCRIPT ORGANIZATION AND FORMATTING:

Sincerely,

Reviewer #2 (Comments to the Authors (Required)):

Manuscript Nr: LSA-2019-00526R
Zelazowska et al., "Gammaherpesvirus Infected Germinal Center Cells Express A Distinct Immunoglobulin Repertoire"

The authors demonstrate that murine gamma herpesvirus 68 (MHV68) infection in mice skews their B cell repertoire. As previously reported for the human gamma herpesvirus Kaposi sarcoma associated herpesvirus (KSHV) MHV68 infected cells are enriched for lambda light chain usage. Furthermore, also their variable region usage in

the BCR heavy chain is skewed, while isotype and somatic hypermutation characteristics do not differ from uninfected germinal center B cells in the same animals. From these data the authors speculate that MHV68 could shape the specificity of their infected B cells, possibly eliciting autoreactivities that could support the survival of their host cells.

In their revised manuscript version, the authors have addressed my previous concerns. They demonstrate that MHV68 does not seem to up-regulate the immunoglobulin recombination and hypermutation machinery and does not seem to favor auto- or virus-reactive B cell populations during its infection. Therefore, the survival experiment (previous major comment #2) would have been very informative. As reviewer #3 points out, it would be beneficial for MHV68 to establish persistence in B cells that provide tonic signaling via their BCR for survival, but do not get activated through viral or self-antigens. Despite this omission, the additional experiments and discussion have significantly improved the manuscript.

Reviewer #3 (Comments to the Authors (Required)):

The finding that γ HV68 subverts GC selection to expand in a specific B cell subset to gain access to long-lived memory B cells is not new to the field: this study's main conclusion. Novel would be the antigenic specificity of the BCR of infected cells, which was not revealed by this study despite the several experimental approaches. I acknowledge the effort made by the authors to address BCR specificity using recombinant antibody methodology and ANA (anti-nuclear antibodies) characteristic for systemic, autoimmune-mediated diseases. This assay failed to give a positive hit. Hence, this study conclusions, after review, are still unchanged but to reinforce that there is no associated autoimmune cells in γ HV68 B cell infection. Nevertheless, the authors continue to favor the message that: "With the establishment of this cloning pipeline, future analysis can address earlier time points and whether there is a connection to self-reactive BCRs during the initial cell infection." I find this unnecessary speculative bias.

2nd Authors' Response to Reviewers January 28, 2020

RE: Life Science Alliance Manuscript - Editorial Decision LSA-2019-00526R

Dear Dr. Leibfried

Thank you for your positive decision to publish our manuscript pending final minor revisions.

You requested the following revisions:

- Please address reviewer #3's remaining concerns via text changes
- Please upload the supplementary tables and include a short legend for them in the main manuscript file

We have made the following changes:

We have removed the text reviewer #3 described as speculative and uploaded the supplementary tables with a short legend in the main manuscript, included in the section with supplemental figure legends.

The FINAL FILES include the above revisions and is submitted as

- An editable version of the final text.
- High-resolution figure, supplementary figures as eps files
- Summary blurb

January 29, 2020

RE: Life Science Alliance Manuscript #LSA-2019-00526RR

Prof. Kevin M McBride
The University of Texas MD Anderson Cancer Center
Epigenetics and Molecular Carcinogenesis
1808 Park Road 1 C
Smithville, TX 78957

Dear Dr. McBride,

Thank you for submitting your Research Article entitled "Gammaherpesvirus Infected Germinal Center Cells Express A Distinct Immunoglobulin Repertoire". It is a pleasure to let you know that your manuscript is now accepted for publication in Life Science Alliance. Congratulations on this interesting work.

DISTRIBUTION OF MATERIALS:

Again, congratulations on a very nice paper. I hope you found the review process to be constructive and are pleased with how the manuscript was handled editorially. We look forward to future exciting submissions from your lab.

Sincerely,

Andrea Leibfried, PhD
Executive Editor
Life Science Alliance

Meyerhofstr. 1
69117 Heidelberg, Germany
t +49 6221 8891 502
e a.leibfried@life-science-alliance.org
www.life-science-alliance.org